# Deconvolution of intergenic polymorphisms determining high expression of Factor H binding protein in meningococcus and their association with invasive disease

Marco Spinsanti[1], Tarcisio Brignoli[1,2¤a], Margherita Bodini[1], Lucia Eleonora Fontana[1], Matteo De Chiara[1¤b], Alessia Biolchi[1], Alessandro Muzzi[1], Vincenzo Scarlato[2], Isabel Delany[1] *

**1** GSK, Siena, Italy, **2** Department of Pharmacy and Biotechnology (FaBiT), University of Bologna, Bologna, Italy

¤a Current address: School of Cellular and Molecular Medicine, University of Bristol, Bristol, United Kingdom
¤b Current address: Université Côte d'Azur, CNRS, INSERM, IRCAN, Nice, France
* isabel.x.delany@gsk.com

## Abstract

*Neisseria meningitidis* is a strictly human pathogen and is the major cause of septicemia and meningitis worldwide. Factor H binding protein (fHbp) is a meningococcal surface-exposed lipoprotein that binds the human Complement factor H allowing the bacterium to evade the host innate immune response. FHbp is also a key antigen in two vaccines against *N. meningitidis* serogroup B. Although the *fHbp* gene is present in most circulating meningococcal strains, level of fHbp expression varies among isolates and has been correlated to differences in promoter sequences upstream of the gene. Here we elucidated the sequence determinants that control fHbp expression in globally circulating strains. We analyzed the upstream *fHbp* intergenic region (fIR) of more than 5800 strains representative of the UK circulating isolates and we identified eleven fIR sequence alleles which represent 88% of meningococcal strains. By engineering isogenic recombinant strains where fHbp expression was under the control of each of the eleven fIR alleles, we confirmed that the fIR sequence determines a specific and distinct level of expression. Moreover, we identified the molecular basis for variation in expression through polymorphisms within key regulatory regions that are known to affect fHbp expression. We experimentally established three expression groups, high–medium–low, that correlated directly with the susceptibility to killing mediated by anti-fHbp antibodies and the ability of the meningococcal strain to survive within human serum. By using this sequence classification and information about the variant, we predicted fHbp expression in the panel of UK strains and we observed that strains with higher expressing fIR alleles are more likely associated with invasive disease. Overall, our findings can contribute to understand and predict vaccine coverage mediated by fHbp as well as to shed light on the role of this virulence factor in determining an invasive phenotype.

**Data Availability Statement:** All relevant data are within the manuscript and its Supporting Information files.

**Funding:** This study was sponsored by GlaxoSmithKline Biologicals SA which had a role in the study design, collection, analysis, interpretation of data and the writing of the manuscript as well as the decision to submit for its publication. This work was supported in part by EUCLIDS Grant FP7 GA 279185.

**Competing interests:** I have read the journal's policy and the authors of this manuscript have the following competing interests: MS and MB were employees of Randstad Italia S.p.A, working as contractors for GSK at the time the study was conducted, and are now employees of the GSK group of companies. LF, AB, AM and ID are employees of the GSK group of companies; MDC was an employee of the GSK group of companies, now at Université Côte d'Azur. ID reports ownership of GSK stocks. ID is listed as inventor on patents on vaccine candidates owned by the GSK group of companies. At the time of the study MS and TB were recipients of a GSK fellowship from the PhD program of the University of Bologna. VS declares no conflicts of interest.

## Author summary

Complement plays a key role in the immunity against *Neisseria meningitidis*. The meningococcus uses the Factor H binding protein (fHbp), to bind a negative regulator of the alternative complement pathway, factor H, to its surface thus preventing complement deposition and lysis. The use of fHbp as an antigen in two licensed vaccines highlights its public health relevance. Therefore the levels of this antigen produced by the bacterium are pivotal on the one hand for the survival of *N. meningitidis* in blood and on the other hand for the susceptibility to vaccine-induced killing antibodies. Here, we identify the predominant nucleotide sequences that drive distinct levels of the fHbp antigen in circulating meningococcal strains. We cluster them into distinct groups with increasing levels and observe that strains expressing higher fHbp amounts are associated with invasive disease. Our findings show that the nucleotide sequence of the *fHbp* promoter can be used for the prediction of antigen levels of any given strain and consequently for both the assessment of its sensitivity to killing by fHbp antibodies and its likelihood to cause invasive disease.

## Introduction

*Neisseria meningitidis* is a strictly human commensal of the naso-pharynx which occasionally can cross the epithelial and blood-brain barriers and cause invasive meningococcal disease (IMD). Carriage rates of meningococcus range from 0.6 to 34% in different global populations [1]; however, IMD is a rare occurrence with rates of <1 to few cases per 100,000 population in areas of sporadic disease to up to 25–100 cases in areas of endemic disease. The disease results in high mortality (up to 10%) and frequent morbidity (up to 20% long term sequelae) [2] (https://www.who.int/en/news-room/fact-sheets/detail/meningococcal-meningitis). Factors determining the establishment of either the development of IMD or the carriage state following acquisition are not completely understood. The reasons for the rare disease manifestation or the switch from colonization to invasion may be multi-factorial and include environmental and genetic bases both on the side of the pathogen and the host. The first evidence that invasive disease may be associated with the genetics of the strain arose when Maiden and colleagues identified hyper-virulent lineages that were overrepresented in invasive isolates and less common in carriage samples [3]. Comparative studies looking at differences in a wide range of carriage and invasive strain genomes have identified a prophage associated with invasive strains in adolescent populations [4,5]. However, major distinguishing genetic differences between carriage and disease isolates have not been identified, suggesting that IMD might be a multi-gene phenomenon [5–7]. There is now strong evidence that host genetic factors influence the occurrence of IMD [8,9] and more recently variations in human Complement factor H (CFH) and related proteins have been found to be important in determining susceptibility to meningococcal disease [10–13].

The complement system plays a key role in human host defense against invasive *N. meningitidis* [14]; indeed, deficiencies in components of the classical, alternative and terminal complement pathways are strongly associated with an increased incidence of IMD [14–16]. Meningococcus has evolved a number of mechanisms to evade complement-mediated killing in the host, including expression of polysaccharide capsules, lipooligosaccharide structure and sialylation [17], and recruitment of the complement down-regulating molecule CFH [18,19].

The surface-exposed lipoprotein Factor H binding protein (fHbp) is a pivotal virulence factor of *Neisseria meningitidis* [20] and, as the name suggests, it specifically binds human CFH [18,21], allowing meningococci to survive and grow in human blood [14,19]. Notably, fHbp is

an immunogenic and protective antigen of the two licensed vaccines against serogroup B meningococcus (MenB) [22,23]. Its coding sequence is highly variable, with more than 1150 different alleles identified so far (http://pubmlst.org/neisseria/fHbp/), and it can be clustered into three variants (var1, var2 and var3) or two subfamilies (A and B), depending on the classification [24,25]. This genetic variation may influence both the fitness of the bacterium in blood and serum when tested in *ex vivo* experiments, and its susceptibility to anti-fHbp antibodies [26–28]. In addition, fHbp expression can significantly vary among strains [24–26,29,30]. Epidemiological studies showed an association between carriage isolates and low levels of fHbp expression [31] and var2 expressing strains have been as well associated more with carriage rather than invasive isolates [32]. Previously, we identified that the promoter region of *fHbp* is highly variable among MenB isolates and we proposed an association between phylogenetic clades of this region and fHbp protein levels [26]. Recently, this association has been further investigated for strains harboring fHbp var1, and specific polymorphisms within the intergenic region sequence have been correlated to determined *fHbp* mRNA levels [33].

Throughout the infection, *N. meningitidis* encounters diverse niches within the host; hence, it is subjected to different environmental conditions, in terms of stress factors and nutrient availability, which influence the expression of virulence factors [34]. Previous studies demonstrated that the *fHbp* gene can be transcribed from a dedicated promoter and by a bicistronic transcript initiating from the promoter of the upstream gene *cbbA* and resulting from read-through of the Rho-independent terminator [30]. Moreover, fHbp expression is regulated by iron availability, oxygen concentration and temperature, and in blood *ex vivo* culture [30,35–38]. In the absence of oxygen, fHbp is upregulated by the global transcriptional factor that regulates the switch to anoxia, namely the Fumarate and Nitrate Reductase (FNR) regulator [30], through binding of the transcription factor and direct activation of the dedicated *fHbp* promoter. Moreover, the FNR-binding site sequence was found to be conserved in the panel of strains analyzed, indicating a conserved oxygen-response regulation among isolates [30]. Loh *et al.* have shown that the increase in temperature results in the increment of fHbp protein amounts and they associated this phenomenon to the presence of secondary structures within the mRNA, called thermosensors, which are composed of the 5' UTR and spanning part of the coding sequence of fHbp (27 nt from the GTG start codon) [36,37]. Indeed, while at low temperatures these structures are tightly formed and limit protein translation, at higher temperatures secondary structures can melt and more protein is translated [37].

To fully understand the polymorphisms that determine different expression of fHbp in diverse *N. meningitidis* strains and that regulate it in response to environmental stimuli, we performed genetic and functional analyses of the region containing all known regulatory sequences of the *fHbp* gene spanning from the stop codon of the upstream gene *cbbA* to 27 nt downstream of the fHbp start codon, here named the *fHbp* Intergenic Region (fIR). This analysis enabled us to identify the polymorphisms important for fHbp expression and hence, to cluster and classify the major fIR alleles into three different expression groups. Interestingly, the prediction of expression levels based on the nucleotide sequence of the intergenic region and variant of *fHbp* in a panel of carriage and invasive isolates collected in the UK enabled us to find an association between those strains predicted to express high levels of fHbp and IMD cases.

## Results

### Identification of the predominant *fHbp* intergenic region (fIR) alleles within *fHbp* promoter clades

In a previous study on a panel of 105 strains selected to be equally representative of the three fHbp variants, we identified eight clades of *fHbp* promoter sequences which were associated

with specific levels of fHbp protein produced in each strain [26]. To understand the molecular basis that determines and regulates fHbp expression, we further analyzed the region, here named _fHbp_ Intergenic Region (fIR), and the genetic variations within and between each promoter clade that may drive the diverse expression levels (Fig 1A). Within each promoter clade we identified the most representative unique sequence or allele, except for clade IV in which two different dominant intergenic region alleles were present, namely fIR3 and fIR6. There are nine dominant fIR alleles that represent 77% of the 105 strains investigated. Five are associated with variant 1 fHbp coding sequences, three are associated with variant 3, while all variant 2 sequences have predominantly one fIR allele, namely fIR4.

The data on quantification of fHbp by Selected Reaction Monitoring-Mass Spectrometry (SRM-MS) [26] were replotted according to the fIR allele clustering (Fig 1B) and while fewer strains might be included, the interquartile ranges (IQR) obtained when plotting fHbp amounts with the respect to either promoter clades [26] or fIR sequence alleles (S1 Table) indicate that the latter analysis results to be more accurate and precise. In Fig 1B we also plotted (grey squares) the strains within the original clade containing polymorphisms with respect to the major fIR allele. Interestingly these are often outliers with respect to their fHbp protein levels, suggesting that some of these polymorphisms may ultimately drive differences in protein expression and the summary of the polymorphisms present within each clade are plotted in S1 Fig. A clear example is the M08-0240104 strain, which contains a polymorphism within the -35 promoter hexamer and exhibits 10-fold higher expression levels than the fIR4 alleles within the same promoter clade. In clade II the strain ISS-2033 exhibiting low expression harbors a SNP within the Rho-independent terminator region with respect to the high expressing fIR1 alleles within the clade and in clade VI a number of outlier higher expressors exhibit SNPs again in the terminator region with respect to the fIR20 containing strains, which have all expression below the lower limit of quantification.

## fIR sequence allele determines distinct _fHbp in vitro_ expression levels in an isogenic background

To assess whether the nine fIR alleles determine distinct _fHbp_ expression levels, we generated a panel of isogenic recombinant strains in the MC58 background where the same fHbp variant (var1.1) was under the control of the nine fIR alleles identified (Fig 2A). The transcript expression levels of _fHbp_ and its upstream gene _cbbA_ in the set of generated mutants were tested by qRT-PCR. Furthermore, the presence of a possible bicistronic transcript was assessed using primers to amplify the intergenic-transcript from the 3' region of _cbbA_ and upstream of the dedicated _fHbp_ promoter transcriptional start. mRNA levels of _cbbA_ were less variable than the _fHbp_ transcript levels between the isogenic strains (Fig 2B vs Fig 2D) and, as expected, no statistical differences between mutants were measured. In the case of fIR alleles 1, 7 and 16 it was possible to detect an intergenic-transcript (Fig 2C), suggesting that a bicistronic transcript is expressed in these strains. These results indicate that the efficiency of the Rho-independent terminator is low in these strains and that read-through from _cbbA_ is possible. The strains harboring fIR1 and fIR7, in which the read-through was observed, together with fIR6 and fIR15 strains showed the highest levels of _fHbp_ transcript (Fig 2D). fIR16 showed intermediate levels of transcript which were statistically lower than fIR7 and the lowest transcript levels of _fHbp_ were detected where transcription was under the control of the fIR alleles 2, 3, 4 and especially 20. Notably, the two polymorphisms that segregate promoter clade IV into fIR alleles 3 and 6 (Fig 2E), appear to result in distinct levels of transcripts. Taken together, these results suggest that the diverse fIR alleles can drive distinct differences in transcript and expression levels of _fHbp_. Given the fact that the genetic background of the recombinant strains is identical, the

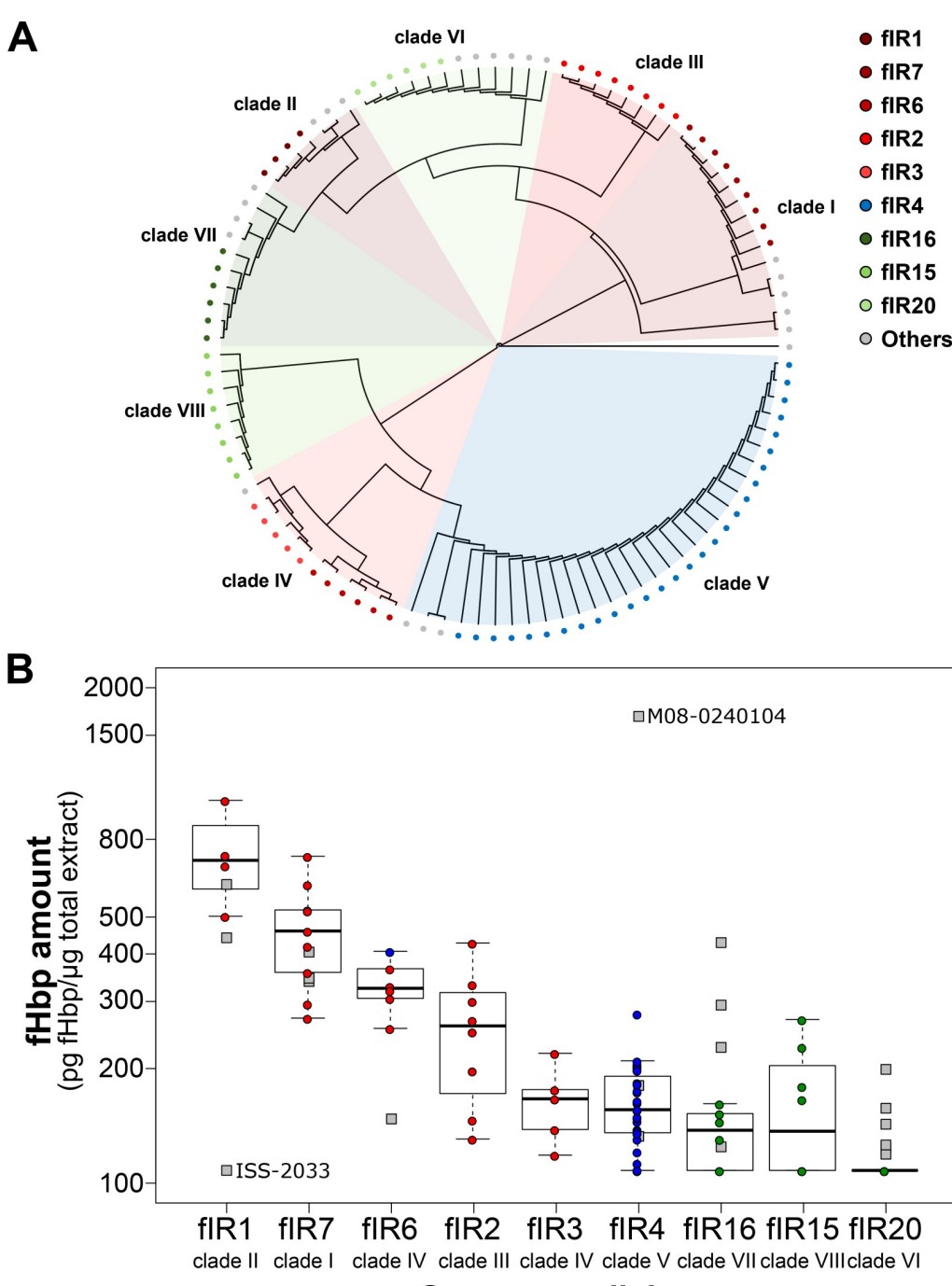

**Fig 1. Alleles of the *cbbA–fHbp* intergenic region (fIR) from 105 strains [26]: variation within promoter clades and resulting expression levels. (A)** Phylogenetic tree obtained from the multiple sequence alignment of the fIR of the 105 strains previously investigated in Biagini *et al*. [26]. Promoter clades from the previous analysis are represented as colored sectors inside the tree: colors of sectors reflect the association with var1, var2, and var3 fHbp as shades of red, blue, and green, respectively. Within each clade the most representative fIR allele(s) are represented as colored dots: shades of red, blue, and green indicating association with var1, var2, and var3 respectively; grey dots indicate fIR alleles other than the most representative. **(B)** Box plots showing the distribution of Selected Reaction Monitoring (SRM)-MS fHbp quantification values of different strains (previously measured and reported in [26]) clustered by major fIR sequence alleles within each clade and ordered from highest to lowest mean fHbp protein values. The Y axis is in logarithmic scale. The median fHbp protein value for each group is indicated by thick bars, the interquartile range is delimited by each box, and the 95% frequency intervals of SRM-MS values are marked by the whiskers. Expression

values of the strains belonging to the clades but not to the fIR alleles (Others) are plotted as grey points and are not considered in the box plots. The names of the outlier strains described in the Results are highlighted.

variability of expression is dependent only on the polymorphisms within the region upstream of *fHbp* affecting differences in termination/read-through or initiation of transcription within the DNA locus or affecting differences in stability or translatability of the RNA.

To elucidate the polymorphisms that might be responsible for the variability in fHbp amounts produced by bacteria, we aligned the sequences of the nine intergenic regions, putting in evidence the regulatory elements therein (Fig 2E). Six fIR alleles maintained the same Rho-independent terminator sequence represented by a 'perfect' palindromic 20 bp sequence followed by 5 Ts [39]; whereas the other three, fIR1, 7 and 16, contained multiple SNPs within one of the palindromic sequences. The free energy predictions of each one of the Rho-independent terminator sequences of each of the intergenic regions [40] indicate that fIR alleles 1, 7 and 16 had low or medium free energy ($\Delta G$ = -13.0; -14.8 and -24.7 kcal/mol, respectively); whereas, the other fIR alleles contained the same strong terminator with $\Delta G$ = -27.3 kcal/mol (S2 Fig). These differences may result in less efficient termination and hence read-through of the RNA polymerase from the upstream gene *cbbA*, as it has been previously reported [30,33]. The sequences of both the FNR binding site and the -35 box were 100% conserved through all fIR sequence alleles, except for the previously mentioned outlier strain M08-0240104. Among the fIR alleles, three different SNPs were identified within the spacer between the -35 and -10 boxes. Moreover, two alleles of the -10 box were found, TACCAT or TACCGC. Interestingly, two out of three sequences containing the TACCGC allele were associated with a weak or medium Rho-independent terminator and therefore read-through from the upstream *cbbA* promoter. Furthermore, an insertion element of 187 bp rich in A and T (previously described as ATR or IE, insertion element) [24,30,35] was identified in the fIR2, 14 nt downstream of the mapped RNA transcriptional start site [30]. The highest degree of variability was observed within the long region described as a putative RNA thermosensor [36]. Notably, a polymorphism (T/C) was found just downstream the ribosome binding site (AGGAG).

## Polymorphisms in the main regulatory regions of *fHbp* drive diverse fHbp expression

To decipher the influence on the fHbp expression levels of the major genetic determinants identified between the different fIR alleles (Fig 2E), we mutagenized the intergenic region of *fHbp* in the MC58 background by site-directed mutagenesis and generated a series of isogenic recombinant strains (Fig 3A) differentiated uniquely by specific polymorphisms. The sequence of the wild-type fIR7 intergenic region was modified with the substitution of a single nucleotide within the stem region of the terminator sufficient to restore a correct base pairing in the stem (position 46 of the alignment, from A to G), increasing its strength from $\Delta G$ = -14.8 to -27.3 kcal/mol, for testing of the 'weak' and 'strong' terminator hypothesis (c*fHbp* and c*fHbp* term, respectively). Both alleles of the -10 box (TACCAT or TACCGC) were generated in either the weak or the strong terminator background. In addition, all four different variants of the spacer region were generated in the promoter with the strong terminator. Finally, in order to investigate the -35 SNP identified in the outlier strain M08-0240104 [26], we generated an isogenic strain with this polymorphism in the strong terminator background. The expression levels of the protein in the set of mutants generated were then tested by Western blot analysis (Fig 3B). When the wild-type terminator (weak) was substituted with the mutated terminator with lower free energy predictions within the stem-loop structure (strong) the amount of

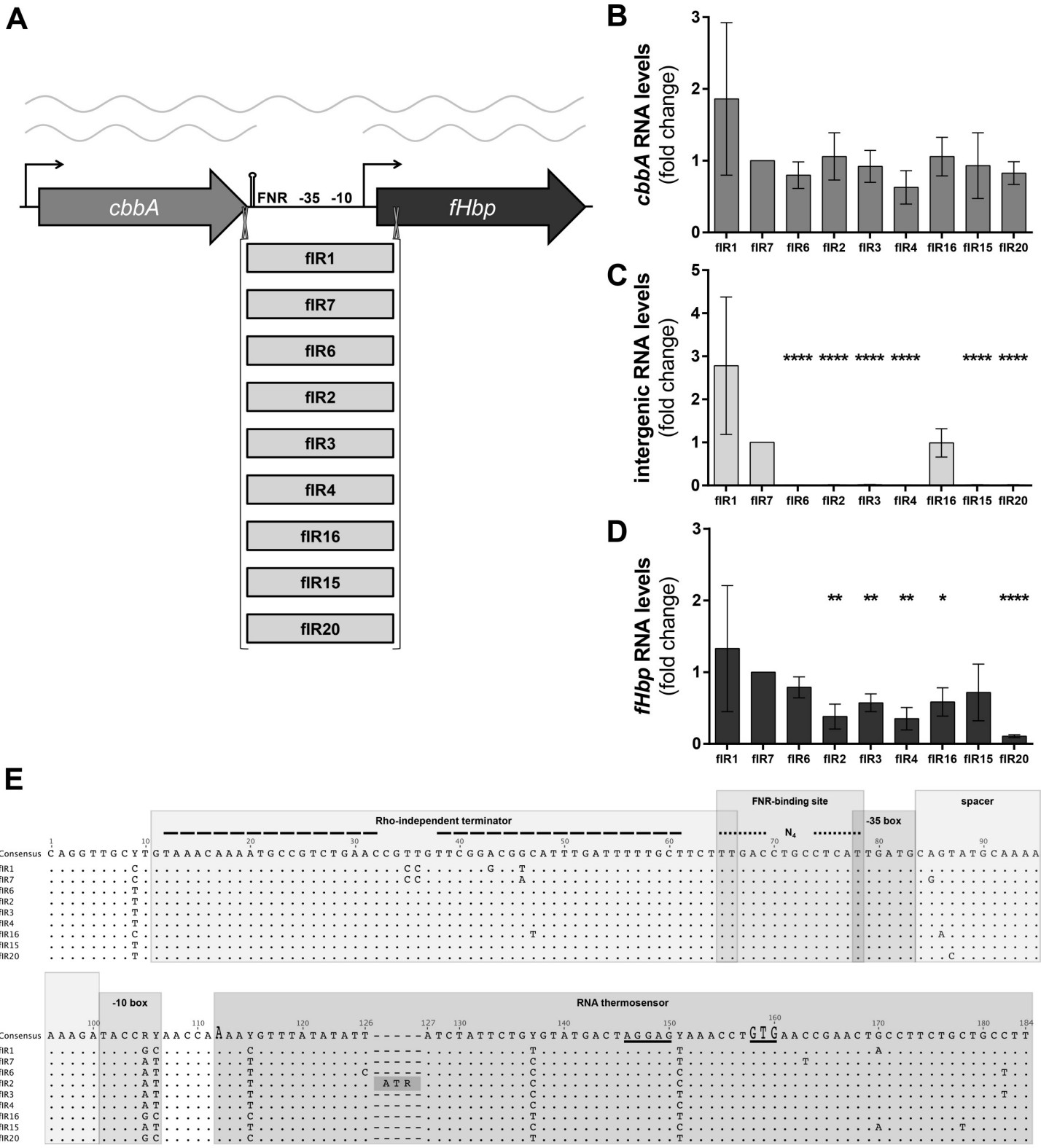

**Fig 2. fIR nucleotide sequence determines fHbp expression levels.** (**A**) Schematic representation of the locus of *fHbp* (dark grey) with its upstream *cbbA* gene (grey) and the transcripts generated from both promoters. The intergenic region (fIR) under investigation and the regulatory elements within it are highlighted: the rho-independent terminator is shown with a stem-loop and the transcriptional start sites of both genes by bent arrows. The set of isogenic recombinant mutants generated in the MC58 background is depicted in light grey. Results of the qRT-PCRs performed on *cbbA* (**B**), the intergenic region (**C**) and *fHbp* (**D**) transcripts. mRNA was

extracted from three independent cultures grown in liquid GC at 37˚C until $OD_{600}$ = 0.5. Values of the single replicates are normalized to the reference genes *16S RNA* and *adk* values within each mutant background, and normalized values are plotted relative to the fIR7 strain. Data represent mean and average of three biological replicates. *p*-values were obtained through Tukey's post-test after one-way ANOVA test: *, $p \leq 0.05$, **, $p \leq 0.01$, ****, $p \leq 0.0001$; no symbol is present when $p > 0.05$. (**E**) Multiple sequence alignment of the nine fIR alleles showing key polymorphisms. The consensus sequence is at the top of the aligned sequences. Dots represent conserved positions and mismatches are indicated with nucleotides. The regions of the regulatory elements are indicated and boxed in shades of grey. Palindromic sequences of the stem of the terminator are dashed. The mapped FNR-binding site is indicated as dotted lines. The insertion sequence rich in A and T (ATR) is boxed. The transcriptional start site is indicated at position 112 and the RBS–AGGAG–and translational start site GTG are underlined.

protein produced substantially decreased (*cfHbp* versus *cfHbp* term). Notably, no differences in the band intensities were observed between the two alleles of the -10 box in the weak terminator-background (*cfHbp* [TACCAT] versus *cfHbp* -10 box [TACCGC]), suggesting that in this scenario read-through overrides the dependency on the dedicated promoter. The strong terminator background allowed the detection of differences in the expression levels of the two -10 box alleles, in that the TACCGC mutant (*cfHbp* term -10 box) produced less fHbp compared to the TACCAT (*cfHbp* term) derivative. By comparing the different alleles of the spacer region, no differences were observed (*cfHbp* term versus *cfHbp* term spacer1, 2 and 3). Interestingly, the aforementioned SNP within the -35 box was responsible for an extensive increase on the expression of the protein (*cfHbp* term -35 box versus *cfHbp* term). Altogether these

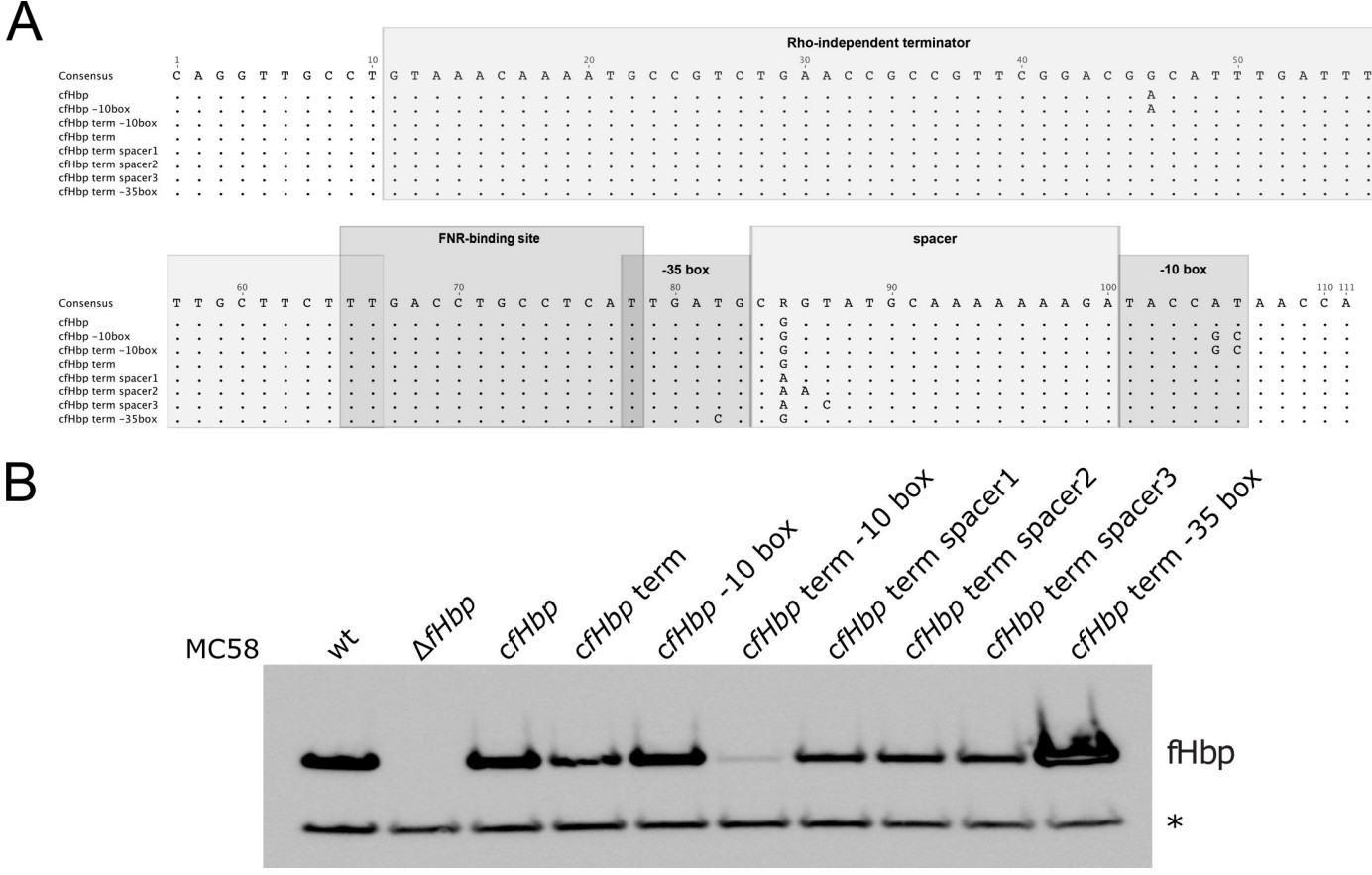

**Fig 3. Polymorphisms that regulate fHbp expression. (A)** Multiple sequence alignment of the MC58 *fHbp* intergenic region (spanning from the translational stop codon of *cbbA* and the transcriptional start site of *fHbp*) and of the mutants of the regulatory elements generated by site-directed mutagenesis. The consensus sequence is at the top of the aligned sequences. Dots represent conserved positions and mismatches are indicated with nucleotides. The regions of the regulatory elements are indicated and boxed in shades of grey. **(B)** Western blot analysis representative of three biological replicates on the set of mutants generated and grown overnight in GC agar at 37˚C. The protein YraP, GNA2091, was used as a loading control and it is indicated with an asterisk.

data indicate that polymorphisms identified in the terminator, the -10 and the -35 elements can have a significant effect on the levels of fHbp from recombinant promoters.

Previous studies showed that secondary structures of the 5' region of the *fHbp* transcript are responsible for the fHbp protein thermoregulation [36,37]. fIR3 and fIR6 share very similar nucleotide sequences except for two polymorphisms within the RNA thermosensor (Fig 2E). We analyzed possible effects of these polymorphisms on the predicted mRNA secondary structure at 30 and 42˚C. While the fIR3 allele is not predicted to change its thermosensor structure at the two temperatures (Fig 4A), the fIR6 structure prediction dramatically changed when shifting from 30 to 42˚C and both were distinct from the fIR3 predicted structure (Fig 4B). Investigation of the protein expression levels from the two strains at distinct temperatures confirmed that the fIR3 strain showed similar fHbp amounts at the two temperatures and significantly less than the fIR6 strain (Fig 4C), which exhibited increased expression at 42˚C compared to 30˚C. Therefore, the predicted changes in the secondary structure of the fIR6 thermosensor in response to temperature seems to be associated with thermoregulation of the fHbp protein, and the stable predicted structure of fIR3 at both temperatures could be associated with overall lower fHbp expression levels. Hence, changes in polymorphisms in the thermosensor of *fHbp* may alter levels and regulation of fHbp and highlights that the thermoregulation mechanism may not be ubiquitous for all fIR alleles.

## Influence of the variant sequence on the expression levels of fHbp

We demonstrated how fHbp expression is influenced by its promoter sequence; however, our analysis was based only on recombinant strains harboring fHbp var1.1. Previous studies have shown that the three fHbp variants might actually have different protein stability [41,42]. To confirm whether fHbp levels may also depend on sequences within the amino acid coding region, we generated a set of recombinant strains where different peptides of fHbp under the control of the same promoter were incorporated in a different locus within the MC58 Δ*fHbp* background (Fig 5A). The transcript expression levels of the different strains were found to be stable indicating similar quantities of mRNA generated (Fig 5B), except for the mutant expressing var1.1, which was significantly higher for unknown reasons and thus was not investigated further. On the other hand, we quantified by SRM-MS the fHbp amounts produced by these strains and we observed that strains expressing var1.1 and var1.14 contained more fHbp than strains carrying var2 and var3 peptides (Fig 5C). Therefore, from the same levels of transcripts, different amounts of protein were generated. Taken together, these data suggest a role of the *fHbp* coding sequence or amino acid sequence either in its translation efficiency or in the stability of the protein respectively, with var1 sequences generally accumulating higher protein levels than var2 and var3 strains, at least for those variants tested.

## Isolates with predicted high fHbp expression show an augmented proportion of invasive meningococcal disease isolates

To extend the analysis to a more significant and wider panel of strains, we extracted and aligned the nucleotide sequence of the *fHbp* gene and its upstream region from 5818 isolates available in the BIGSdb [43]. These strains represent isolates of meningococcus collected in the UK both from carriage and from IMD before 2016. The panel contains strains belonging to different serogroups with a majority of serogroup B (45%, S3A Fig), and to several CCs, including those hyper-invasive (S3B Fig). A total of 171 different fIR alleles were identified; in addition to the nine fIR alleles identified in the 105 strains, the two new most frequent alleles were fIR11 and fIR13, and these 11 fIRs represented 88% of the strains. Interestingly, fIR11 is similar to fIR7, except for polymorphisms within the terminator region and in positions +90

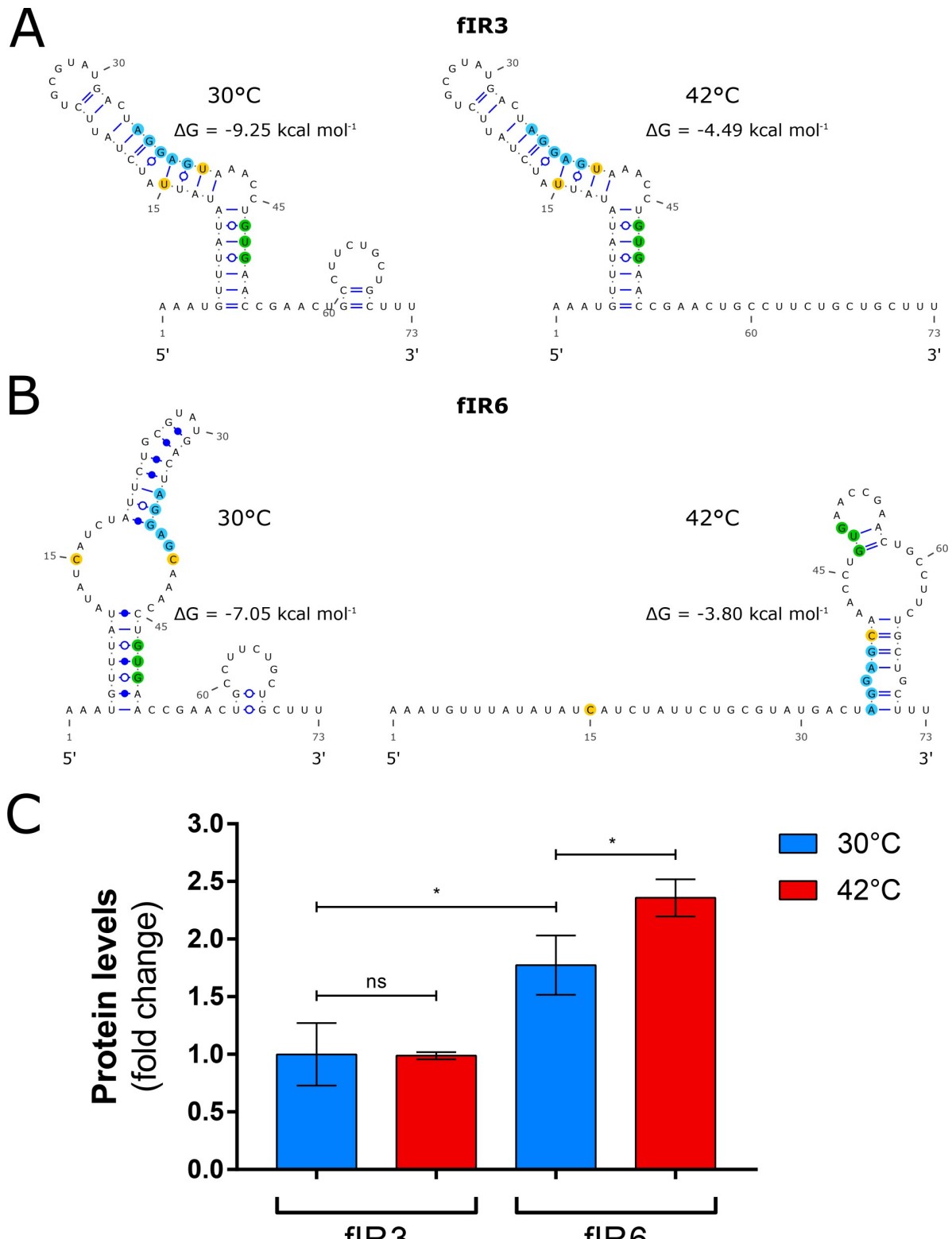

**Fig 4. fHbp thermoregulation is not ubiquitous.** Schematic representation of the RNA secondary structures of fIR3 (**A**) and fIR6 (**B**) at 30˚C (left panels) and 42˚C (right panels). Light blue circles highlight the Shine Dalgarno sequence (RBS); green circles the start codon of fHbp; and yellow circles the polymorphisms that differentiate the two intergenic region alleles. ΔG values of the mRNA secondary structures at each temperatures are indicated. (**C**) Protein levels of fHbp measured at 30˚C (blue) and 42˚C (red) for fIR3 and fIR6 mutants. fHbp is

detected with a polyclonal serum raised against var1.1. Values derived from three biological replicates and statistical significance is calculated with the Student's t test.

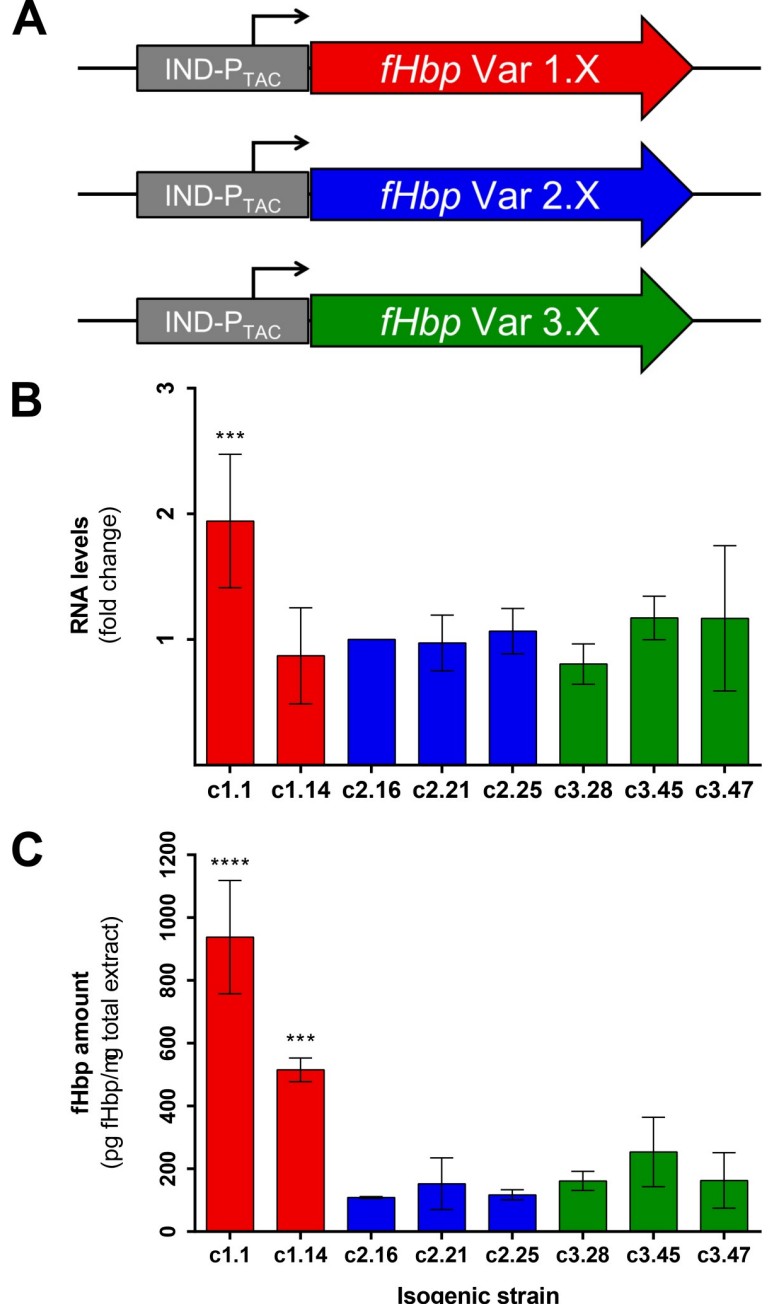

**Fig 5. Influence of the variant sequence on the expression levels of fHbp.** **(A)** Schematic representation of the isogenic recombinant mutants generated. The IPTG-inducible promoter is colored in grey. fHbp peptides are colored in red, blue or green according to their variant group 1, 2 or 3, respectively. Results on the expression levels of fHbp in three biological replicates as measured by qRT-PCR **(B)** and SRM-MS fHbp protein quantification analyses **(C)** are reported. Values on the RNA levels are normalized to the reference genes *16S RNA* and *adk* and to the c2.16 strain. *p*-values were obtained through Tukey's post-test after one-way ANOVA test.

(spacer), +113 and +120 (RNA thermosensor) of the intergenic region; whereas, fIR13 differs from fIR6 only for a SNP at position +126 of the intergenic region (S2 Table). The fIR isogenic mutants in MC58 expressing var1.1 under the control of fIR11 and fIR13 were generated and the respective expression levels, along with the other nine mutants, were tested by Western blot, to measure overall ability to drive protein expression (Fig 6A and 6B). By considering the expression levels it was possible to identify three different groups, high expressors, containing fIR alleles 1 and 7, medium (fIR6, fIR11 and fIR16) and low expressors (fIR alleles 2, 3, 13, 15, 4 and 20).

To assess the capability of antibodies raised against fHbp var1.1 to mediate killing of strains expressing different amounts of the antigen by natural fIR alleles, we performed a serum bactericidal assay on representatives from each group (Fig 6C). Interestingly, by comparing the SBA titers obtained it was possible to group the strains into the same categories identified by Western blot. Furthermore, by correlating the SBA titers with fHbp expression levels we determined a Pearson correlation coefficient of 0.96 (*p*-value: 0.0002).

We then tested the same set of mutants for their capacity of surviving in human serum (Fig 6D). Percentage of survival after three hours of incubation was evaluated by CFU counting and, despite some discrepancies in the identified grouping, high and medium fHbp expressors were recovered at higher percentages than low expressors. The greatest discrepancy was the fIR20 strain, which exhibits the lowest *in vitro* expression profile but survived in serum similar to the medium expressors. These results could suggest 1) a possible regulation of *fHbp* in the presence of human serum for at least some of the fIR alleles under investigation or alternatively 2) that other additional factors may be responsible for the serum resistance in these strains outside of the *fHbp* locus [17,38,44,45] and this was not investigated further. Therefore, fHbp expression levels, determined by natural promoters from an *in vitro* growth culture, correlated well with antibody- and complement-mediated killing of the isogenic strains carrying the diverse panel of fIR alleles.

In a previous study the fHbp expression has been reported to be linked to clonal complexes (CCs) [35], however our data suggest that rather than correlating with any of the conventional classifications such as sequence type, clonal complex or serotype [26], the expression level within each strain correlates to and is determined by the intergenic region upstream of *fHbp* and therefore the fIR allele. We looked for associations between fIR alleles and fHbp variants (S4 Fig), peptides (S5 Fig) and clonal complexes (S6 Fig) and results are summarized in Table 1. We found a strong association between fIR alleles and fHbp peptides, pointing towards a coevolution of the two sequences, which is in line with their linkage. Furthermore, often fIR allele and peptide pairs are associated with a specific clonal complex as it was previously reported [46]. There were however some interesting exceptions to this general rule. Firstly, it is intriguing to have found the presence of intergenic regions that are associated with specific clonal complexes expressing different fHbp peptides, suggesting both a high recombination rate of the downstream coding sequence and a selective pressure to maintain the same regulatory elements for fHbp expression. Secondly, we found a strong association between nearly all peptides of var2 expressed from diverse clonal complexes (ST-23, ST-11, ST-41/44, ST-22, ST-269, ST-167, ST-35, ST-1117, ST-103, ST-162, ST-174 complexes) and the same low expressing-fIR4 allele. Interestingly, while fIR6 is associated with the ST-41/44 complex, the fIR3 allele is found in distinct multiple complexes (ST-1157, ST-269, ST-60, ST-174, ST-213 complexes) once again pointing to a divergent evolution of these sequences as the functional data suggest. Moreover, for each of the clonal complexes ST-11, ST-41/44, ST-269, ST-32, and ST-213, it is clear that there is a larger proportion of higher expressing fIRs associated with the invasive rather than carriage isolates (S6B Fig).

Finally, using the fIR-based analysis and taking into account the greater expression of var1 when compared to var2/3 strains, we divided this panel of strains into four groups, namely

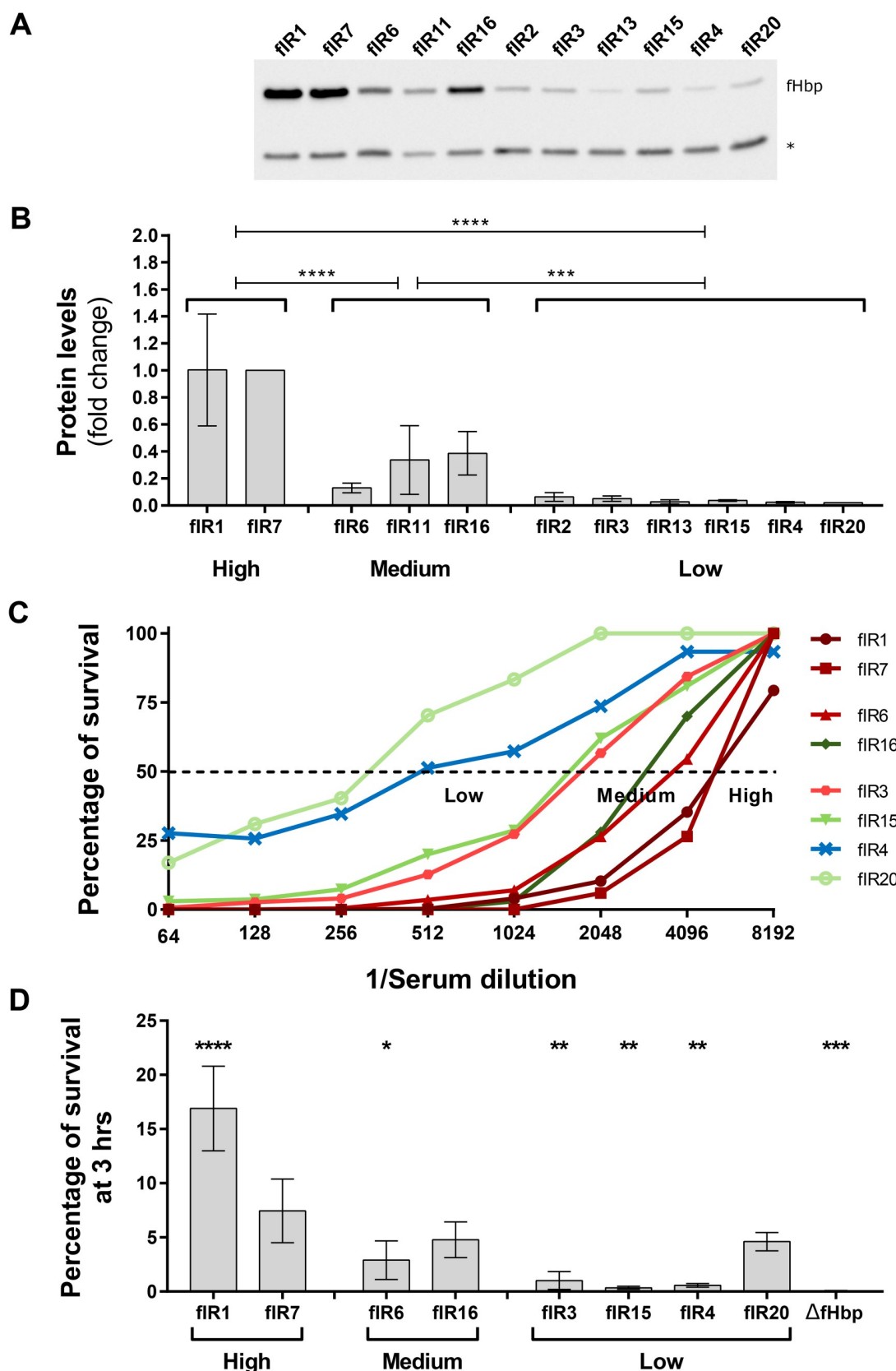

**Fig 6. fIR alleles cluster into three expression groups which are correlated with bactericidal killing and serum resistance.** **(A)** Western blot analysis of the fIR allele mutants grown in MH + 0.25% glucose at 37˚C. The protein YraP, GNA2091, was used as a loading control and it is indicated with an asterisk. **(B)** fHbp protein levels obtained with the quantification of bands intensities. Data represent mean and average of three biological replicates. **(C)** Results of the SBA analysis done incubating bacteria in human plasma as source of exogenous complement and serial dilutions of antibodies against fHbp var1.1. For each strain percentages of survival are plotted according to the inverse of the serum dilution. The dashed line indicates the half of the survival from which SBA titers are obtained. **(D)** Percentages of survival after 3 hours incubation in 57% human serum were calculated with the respect of T0. Data represent mean and standard deviation of three biological replicates. *p*-values were obtained through Tukey's post-test after one-way ANOVA test.

high (fIR1 and 7, 7.7%), medium (fIR6, 11 and 16, 8.2%) and two low groups segregated based on the variant they express, low var1 (fIR2, 3 and 13, 27.7%), and low var2/3 (fIR4, 15 and 20, 56.5%). In order to understand whether fHbp predicted expression levels were correlated with the extent of disease status, either carriage or IMD, we performed a logistic regression on this dataset. In Fig 7 red lines are the 1000 repetitions of the regression done on balanced bootstrap subsets of the public database to overcome the possible bias introduced by an unbalanced dataset. For each expression group, we reported the median percentage of strains that were recovered from cases of IMD (bar plot in Fig 7). Interestingly, the expression levels were significantly predictive of the invasive disease by the ANOVA test, with a median *p*-value of $2.3 \times 10^{-86}$ (ranging from $1.5 \times 10^{-98}$ to $2.0 \times 10^{-76}$), reflecting a positive trend in the proportion of invasive strains with higher predicted expression levels for fHbp. The model does not sufficiently explain the dependent variable, namely the disease status, as uncovered by the McFadden index (median value of 0.066, ranging from 0.058 to 0.075). Therefore, with our analysis we detected an augmented proportion of strains predicted to express high levels of fHbp in patients with IMD.

## Discussion

Factor H binding protein (fHbp) is an important virulence factor of *Neisseria meningitidis* and a component of two licensed vaccines against MenB: 4CMenB (*Bexsero*, which contains subvariant 1.1), and bivalent rLP2086 (*Trumenba*, which contains subvariant 1.55 -subfamily B-

**Table 1. Association between fIR alleles and fHbp peptides and clonal complexes.**

| fIR allele | Expression group | fHbp variant | fHbp peptide | Clonal complex | igr_up_ NEIS0349 | igr_up_ NEIS0350 | fHbp IGR cluster |
|---|---|---|---|---|---|---|---|
| fIR1 | High | 1 | 15 | ST-269 complex | 4 | 2 | E5 |
| fIR7 | High | 1 | 1 | ST-32 complex | 6 | 3, 7 | E4 |
| fIR6 | Medium | 1 | 14 | ST-41/44 complex | 7 | 1 | E2 |
| fIR11 | Medium | 1 | variable | ST-11 complex | 5 | 1 | E3 |
| fIR2 | Low | 1 | 4 | ST-41/44 complex | 2 | 1, 79, 6 | E1 |
| fIR3 | Low | 1 | 13 | variable | 3 | 57 | E2 |
| fIR13 | Low | 1 | variable | variable | 1 | 57 | E2 |
| fIR4 | Low | 2 | variable | variable | 1 | 1, 4 | E2 |
| fIR16 | Medium | 3 | variable | ST-213 complex | 9 | 5, 77 | NA |
| fIR15 | Low | 3 | 47 | ST-461 complex | 1 | 57 | E2 |
| fIR20 | Low | 3 | 102, 45 | ST-53 complex, ST-213 complex | 8 | 77, 10 | NA |

For every fIR allele defined in our analysis, we report the expression group dividing the fIRs in three levels of expression (low, medium and high), and the most frequently associated fHbp variant, fHbp peptide, clonal complex and igr_up_NEIS0350, as reported in S4–S7 Figs, respectively. Furthermore, it is reported the correspondent public allele (igr_up_NEIS0349) as it has been defined in the public database PubMLST [43] and the IGR cluster of expression they have been assigned to [33].

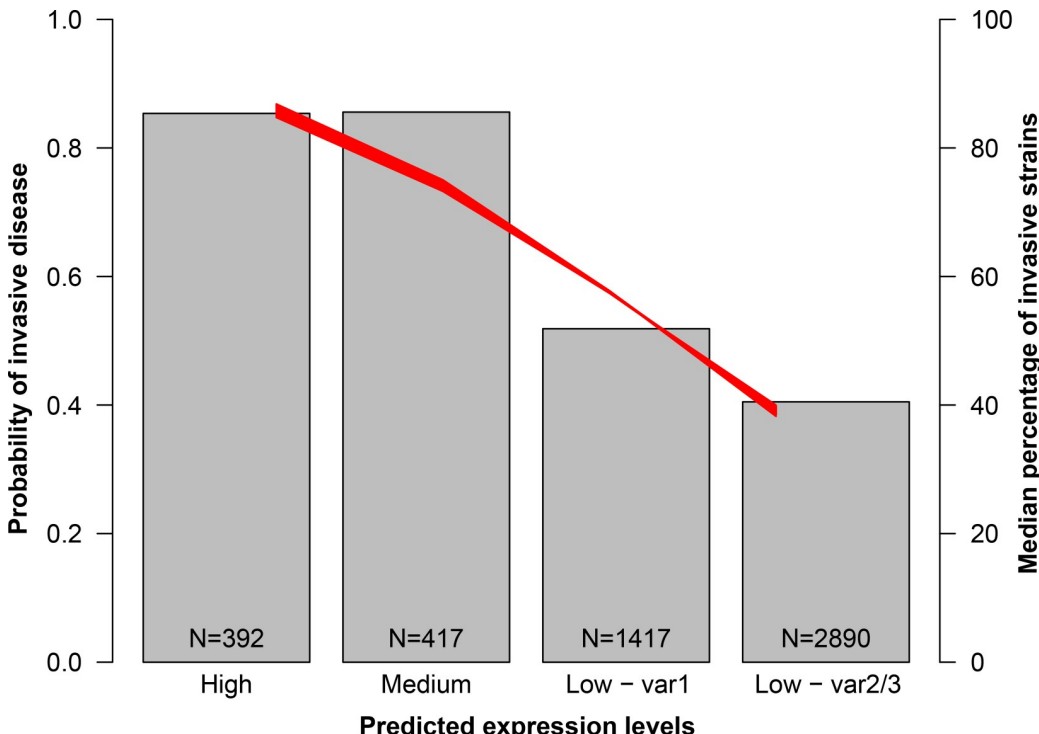

**Fig 7. High fHbp levels correspond to an augmented fraction of invasive disease isolates in a public collection.** The bar plot (right Y axis) reports the proportion of strains isolated from IMD for each predicted-fHbp expression group, calculated as the median of the 1000 balanced resampling. The red lines refer to the left Y axis and are the 1000 logistic regressions performed on the datasets which were balanced for the invasive and carriage isolates contained. On the left Y axis, 0.0 is carriage and 1.0 is IMD. The models explain the carriage/invasive state by means of the predicted levels of fHbp.

and 3.45 -subfamily A-). The key role of fHbp [20,28] in binding CFH and therefore in resistance of meningococcus to complement is highlighted by the fact that nearly all strains express this antigen [29,30,46,47]. Expression of fHbp, and in particular its expression level [26,30], is paramount both for the survival of meningococcus in the blood [38] and for the achievement of complement-mediated bactericidal killing of a given strain by anti-fHbp antibodies [24,26]. In this study we investigated the reasons for the diverse expression levels between circulating meningococcus strains and their implications for both meningococcal virulence and vaccine coverage. Here, building on a previous study where common sequences within the *fHbp* intergenic region (fIR) were noted to determine distinct levels of fHbp expression [26], we elucidate the main functional polymorphisms driving fHbp expression. A molecular analysis of the *fHbp* intergenic regions revealed a major contribution of polymorphisms in the Rho-independent terminator, as described previously [30,33], and the -10 and -35 boxes of the dedicated *fHbp* promoter on the antigen expression levels. Additional contribution from SNPs in the thermo-regulatory region and in the coding sequence *per se* was shown to affect expression levels. Highest expression is achieved when fHbp is driven by both *cbbA* and *fHbp* promoters when polymorphisms weaken the terminator downstream of *cbbA*, or through the outlier SNP in the -35 region, here identified in only 12 out of over 5800 sequences of this study. This mutation in the -35 box may render the promoter less dependent on FNR activation and therefore with higher transcriptional initiation under aerobic conditions, resulting in the increases measured. The presence of a bicistronic transcript deriving from the upstream P$_{cbbA}$ has been previously observed for MC58 [30] and more in general for isolates belonging to ST-32 [35] or ST-269

[33]. Here we experimentally determined that it can be ascribed to fIR1, 7 and 16 strains with intergenic regions having weak terminators which are strongly associated with clonal complexes ST-269, ST-32 and ST-213, respectively. Interestingly, strains in these clonal complexes are generally hyper-virulent and while fIR1 and fIR7 are linked to variant 1 fHbp, fIR16 drives variant 3 expression. The most frequent -10 hexamer for the *fHbp* promoter is TACCAT, but several strains contain the TACCGC variant which diverges significantly from the consensus -10 sequence. We show that promoters with the GC variant express significantly less than the TACCAT hexamer. Interestingly, only three fIR alleles contain the TACCGC allele and two of them (fIR1 and fIR16) are associated with weak or medium terminators and read-through from *cbbA* that would counterbalance its negative effect on fHbp expression. Cayrou and colleagues investigated polymorphisms in the upstream *cbbA* promoter and observed only one allele, allele 6, with significantly lower expression [33]. This is in line with previous observations that the expression of *cbbA* in 41 distinct strains was largely stable, when compared to the magnitude of diverse fHbp expression levels [30]. In addition, strains where fHbp may be expressed also from the *cbbA* promoter through read-through due to fIR1, 7 or 16 alleles were found not associated with the *cbbA* promoter allele 6 (Table 1 and S7 Fig) and carry alleles that have been reported to have similar expression levels [33]; therefore, the distinct fHbp expression levels are likely driven through fIR intergenic polymorphisms alone.

Recently, fHbp has been shown to be thermoregulated [36]. In particular, two copies of anti-RBS hexameric sequence within the CDS of fHbp were identified as potentially able to pair with the ribosome binding site (RBS) hence hindering the access of the ribosome to the *fHbp* transcript at low temperatures [37]. In our analysis we found polymorphisms in both anti-RBS repeats, which could affect base-pairing to the RBS suggesting that thermoregulation of fHbp from the proposed mechanism may vary between strains. Two intergenic regions that we tested for thermoregulation, fIR6 and fIR3, differ only in nucleotides upstream of the start codon and these differences predicted diverse secondary structures of the transcript at the two temperatures tested, which agreed with the presence or absence of thermoregulation of the fIR6 and fIR3 strains respectively, both which drive variant 1 expression in circulating strains. Hence, our data suggest that not all fIR alleles undergo thermoregulation in *N. meningitidis* and that this phenomenon may occur in distinct strains due to alternative anti-RBS sequences to those previously identified [37] and we propose that alternate structures resulting from polymorphisms in this region may also occlude the RBS driving decreased steady state levels and differential expression levels, similar to what has been described by Bhattacharyya and colleagues [48]. This indicates that polymorphisms may affect fHbp amounts at a post-transcriptional level in addition to those SNPs affecting transcriptional initiation or termination. Moreover, other post-transcriptional effects can result from the coding sequence (CDS) of fHbp. Less protein levels of the var2 and var3 could be measured with the respect of var1 when driven from similar promoters, suggesting that the CDS influences either translation efficiency or the amino acid sequence influences protein stability. The variant 3 and in particular variant 2 sequences were reported to have low thermal stability for the amino- (N-) terminal domain compared to variant 1 [42] which may affect accumulation of the protein and therefore expression levels. In a previous study we identified isolates harboring variant 1 as having significantly higher expression levels than those with variant 2 and 3 [26]. Here we show that the reasons for this are multiple: firstly, var1 strains are generally associated with higher expressing fIR alleles than var2 and var3 and secondly, var1 protein sequences tend to accumulate to higher levels and as such ultimately the fHbp protein amounts in natural circulating strains would be influenced by both fIR sequence and CDS.

Cayrou and colleagues recently reported that mRNA level analysis of *fHbp* from nearly 80 strains could cluster var1-expressing strains into five expression groups based on their

intergenic region sequences [33]. Here, through analysis of nearly 6000 meningococcal isolates, we found a relatively small number of fIR alleles which commonly control fHbp expression (11 alleles out of 171 covering 88% isolates) including isolates of var2 and 3, and define major expression groups based on experimental data. Combining the fIR-based and variant-based analyses we divided a panel of strains collected in the UK into four groups based on predicted levels (high–medium–low var1 –low var2/3) of fHbp. The panel analyzed contained carriage and invasive disease isolates, and the 11 common fIRs were all found in both carriage and invasive groups. Nevertheless, even though we cannot predict the disease status solely on the inferred fHbp expression levels, we detected an augmented proportion of isolates with predicted high fHbp expression in IMD isolates. The association of high fHbp expression with IMD isolates is in agreement with the criticality of fHbp for meningococcal survival in blood [18,28]. We further demonstrate here that alteration of fIR in an isogenic strain is adequate to drive different protective outcomes of the meningococcus strain after incubation in serum. Higher fHbp would increase the CFH bound to the bacterium and downregulate complement activation on the surface, hence reducing complement-mediated killing. Other polymorphisms affecting increased capsule expression and generic ability to resist complement have been previously reported in invasive isolates and have been associated with increased resistance to bactericidal killing in strains causing IMD [49]. Polymorphisms within the CFH-fHbp interaction on both the side of the host within the *CFH* locus [10,11,13,50], and the pathogen in the *fHbp* locus [51,52] have been previously associated with disease onset or outcome. Variance in meningococcal fHbp sequences has been previously shown to be associated with disease severity. Piet and colleagues have identified the D184 SNP in the coding region which when present in the fHbp of an invasive strain influences the clinical course of meningococcal meningitis and is associated in a higher rate of unfavorable outcome [51]. In a separate study, patients affected by *N. meningitidis* with fHbp subfamily A were significantly younger than those who had infection with *N. meningitidis* with fHbp subfamily B [52]. From the side of the host, polymorphisms in the *CFH* locus have been correlated to increased susceptibility to invasive meningococcal disease [10]. It would be interesting to investigate the presence of any combination of polymorphisms in the *CFH* region of the host and polymorphisms in the intergenic region of *fHbp* and whether this might somehow be one of the causes for the development of IMD rather than carriage.

Given the strong association of fIR alleles with fHbp peptides and clonal complexes, it becomes complicated to evaluate the specific impact of each variable in determining disease onset in natural strains. The expression groups used in the analysis were determined experimentally and contained more than one fIR allele; in our model these groups were considered as ordinal variables and the probability of a strain to be associated with IMD increases as the predicted fHbp expression level increases. Therefore, these characteristics confidently suggest that fHbp expression levels predicted using fIR sequences play a relevant role in the disease.

The strong correlation between nearly all peptides of var2 expressed from diverse clonal complexes (ST-23, ST-11, ST-41/44, ST-22, ST-269, ST-167, ST-35, ST-1117, ST-103, ST-162, ST-174, ST-213, ST-32) and the same low producing-fIR4 allele is a noteworthy observation from this study and sheds light on the classification of fHbp variants which has been somewhat controversial. Either 3 variants [24] or two subfamilies [25] are considered, and whether var2 and var3 could be classified as one subfamily is a fundamental molecular epidemiological conundrum. This study shows that apart from the genetic distance of the coding sequences, there is a fundamental difference between var2 and var3 with respect to their associated regulatory regions. While almost all variant 2 strains have the fIR4 allele, var3 are associated with remarkable diversity at the promoter level and this observation stresses their segregation into two groups [24] rather than a single subfamily [25].

Interestingly, the grouping of three distinct fHbp levels corresponded intuitively with three groupings of correlated susceptibility to anti-fHbp antibodies bactericidal killing. These findings could add value and information to genotyping tools that predict vaccine strain coverage, such as the Genetic Meningococcal Antigen Typing System (gMATS) [53], which is based on the relationship between protein sequence of antigens and MATS coverage predictions and take into account their expression levels only indirectly. In fact, MATS estimations are based on the result of a sandwich enzyme-linked immunosorbent assay (ELISA) whose signal is determined by the convolution of antigen molecular distance from the vaccine antigen and its expression level. Knowledge on the molecular mechanisms that drive and regulate fHbp expression in meningococcus would allow the deconvolution of expression and molecular variability in MATS measurements. This will reflect in the improvement of gMATS predictions and might help reducing underestimation of protection, which is mainly determined by the appearance in meningococcal disease cases of fHbp molecular subvariants whose expression is completely uncharacterized. Due to the strong association between fHbp peptides and fIR alleles, in terms of general capability to predict strain coverage the two loci are bringing overlapping information. With this work a further refinement on prediction capabilities may be possible, in order to address cases of recombination which result in new intergenic/coding sequence combinations.

The evidences presented here point towards the use of the nucleotide sequence of the *fHbp* intergenic region as a predictive tool for the amount of antigen produced by any strain. This is particularly relevant for the cases where the causative strain is not recovered and occurs in almost 30–50% of cases of meningococcal disease [54]. The regulation of fHbp expression is highly complex and in this study we have limited the use of sequence-based prediction to those fIR for which we have experimental data. The sequence alone can ultimately be informative on the likelihood of the strain to progress to IMD or to be killed by anti-fHbp antibodies that may be induced through vaccination; however, due to the diversity in sequence of the *fHbp* locus there may be some strains for which no precise information through the sequence can be inferred.

## Materials and methods

### Ethics statement

Human serum used in the study was commercially available; while human plasma was obtained according to Good Clinical Practice in accordance with the declaration of Helsinki and patients have given their written consent for the use of the samples of study MENB REC 2ND GEN-074 (V72_92). The study was approved by the Western Institutional Review Board (WIRB).

### Bacterial strains and culture conditions

*Neisseria meningitidis* strains used in this study are reported in S3 Table. Strains were routinely cultured overnight on Gonococcus (GC) agar medium (Difco) with Kellogg's supplement I [55] at 37˚C in an atmosphere of 5% $CO_2$. Liquid cultures were grown in GC with Kellogg's supplements I and II or in Mueller Hinton (MH) broth plus 0.25% glucose at 180 rpm, at 37˚C in an atmosphere of 5% $CO_2$. When required, erythromycin (5 µg/ml), chloramphenicol (5 µg/ml) or isopropylβ-D-1-thiogalactopyranoside (IPTG) (1 mM) (Sigma) were added to culture media at the indicated final concentrations. Thermoregulation was tested by growing the strains till $OD_{600} = 0.5$ at 30˚C, then splitting and incubating them at either at 30˚C or at 42˚C for 30 min. *Escherichia coli* DH5α and Mach1 (Thermo Fisher Scientific) strains were grown in Luria-Bertani (LB) medium, and when required, ampicillin or chloramphenicol were added to achieve a final concentration of 100 µg/ml and 10 µg/ml, respectively.

## Construction of mutant and complementation strains

DNA manipulations were carried out routinely as described for standard laboratory methods [56]. The plasmid pBS-c741 wt CmR (S5 Table), a derivative of the plasmid pBS-c741 wt KanR used in [30], which contains the promoter and the coding sequence of *fHbp* of the MC58 strain, the *cat* gene and a downstream region for the homologous recombination, was used as a template for the generation of both fIR allele mutant plasmids and plasmids of the mutants of the regulatory elements of the intergenic region of *fHbp*. The generation of the mutant plasmids where the expression of fHbp var1.1 was under the control of the different fIR alleles was performed by polymerase incomplete primer extension (PIPE) method [57] using primers vPCRpBSc741-F/R for the amplification of the backbone plasmid pBS-c741 wt CmR and primers iPCRprom-F/R for the amplification of the intergenic regions (S4 Table). Plasmids of the mutants of the main regulatory regions were generated by site-directed mutagenesis carried out using primers TACCAC_TACCGC-F/R (pBS-c741 PfHbp -10 box (TACCGC) CmR), GACGACA_GACGGCA-F/R (pBS-c741 PfHbp term -27 CmR or pBS-c741 PfHbp term -27, -10 box (TACCGC) CmR, from weak to strong terminator), CGGTATG_CAGTATG-F/R (pBS-c741 PfHbp term -27, spacer1 CmR), CAGTATG_CAATATG-F/R (pBS-c741 PfHbp term -27, spacer2 CmR, using spacer 1 as template), CAGTATG_CAGCATG-F/R (pBS-c741 PfHbp term -27, spacer3 CmR, using spacer 1 as template), and TTGATG_TTGACG-F/R (pBS-c741 PfHbp term -27, -35 box (TTGACG) CmR). The *ex locus* complementation of the IPTG-inducible fHbp in the region between the converging ORF NMB1428 and NMB1429 was carried out using the *fHbp* gene amplified (using primers 741-F2/R2 for var1.1, EP1For1.1 and EP5RV1.4 for var1.14, EP2For1.4 and EP6RV2.1 for var2.16 and var2.25, EP1For1.1 and EP6RV2.1 for var3.28 [58], 2.21 fw/rev for var2.21, 3.45 fw and 3.45/47 rev for var3.45, and 3.47 fw and 3.45/47 rev for var3.47) and cloned as a NdeI-NsiI fragment into the pComP$_{IND}$ plasmid [59]. All PCR amplifications were performed using the KAPA Hi-FI polymerase (KAPA Biosystems) and digesting the DNA template with the DpnI enzyme when required. The correct nucleotide sequence of each plasmid was confirmed by DNA sequencing. The plasmids were linearized and used for the transformation of the MC58 Δ*fHbp* strain to create complementation mutants (S3 Table) that were selected on GC agar with chloramphenicol. All transformants were verified both by Western blot and by PCR analysis for the correct insertion by a double homologous recombination event (pairs of primers pRTNM_nmb1869U-F/ pRTfHbpU.R and CmR-down/complcheck-dsGENOME-R, COM-C-Fw/CM-UP-C and pRTfHbpU.F/COM-C-Rev were used for the *in locus* or *ex locus* complementation, respectively).

## RNA isolation and cDNA preparation

Bacterial cultures were grown in liquid medium to an OD$_{600}$ of 0.5–0.6 and then added to 3 ml of frozen medium to bring the temperature immediately to 4˚C. Cells were harvested by centrifugation at 3400 *x g* for 10 minutes. Total RNA was isolated using the RNeasy Mini kit (Qiagen) following manufacturer's instructions. RNA samples were incubated with RQ1 RNase-Free DNase (Promega) for an hour at 37˚C and purified with the RNeasy Mini kit. 2 μg of total RNA were reverse-transcribed using random hexamer primers and SuperScript II RT (Thermo Fisher Scientific) following manufacturer's instructions.

## Quantitative real-time PCR (qRT-PCR) experiments

Quantitative real time-PCR was performed with triplicate biological samples in a 25 μl reaction mixture containing 2.5 ng of cDNA, 2X Platinum SYBR Green qPCR SuperMix-UDG with Rox (Thermo Fisher Scientific) and 0.4 μM of gene-specific primers (S4 Table). Amplification

and detection of specific products were performed with a Mx3000P Real-Time PCR system (Stratagene) using the following procedure: 95˚C for 10 min, followed by 40 cycles of 95˚C for 30 s, 55˚C for 30 s and 72˚C for 30 s then ending with a dissociation curve analysis. The *16S RNA* and *adk* genes were used as endogenous reference controls and the relative transcript change was determined using the $2^{-\Delta\Delta Ct}$ relative quantification method [60]. Data were analyzed with the Graphpad Prism 7 software and statistical significance ($p < 0.05$) was calculated through Tukey's post-test after one-way ANOVA test.

## Western blot analysis

Strains grown overnight on GC agar plates were resuspended in PBS to an $OD_{600}$ of 0.80. One milliliter of the resuspension was centrifuged for 5 min at 15000 *x g* and the pellet was resuspended in 160 μl of SDS loading buffer (50 mM Tris-HCl [pH 6.8], 2.5% SDS, 0.1% bromophenol blue, 10% glycerol, 5% β-mercaptoethanol, 50 mM DTT) [30]. In the case of liquid cultures, strains were grown till an $OD_{600}$ of 0.50 and one milliliter of the culture was pelleted and resuspended in 100 μl of SDS loading buffer. Protein extracts were separated by SDS-PAGE on NuPAGE Novex 4–12% Bis-Tris Protein Gels in MES 1X (Life Technologies) and then transferred to nitrocellulose membranes. Membranes were blocked overnight at 4˚C with PBS + 0.05% Tween 20 (Sigma) and 10% powdered milk (Sigma). A mouse serum anti-fHbp var1.1 fused to the YraP protein was diluted (1:2000) in PBS + 0.05% Tween 20 and 3% powdered milk and incubated for 1 h with the membrane. A horseradish peroxidase-conjugated anti-mouse IgG antibody and the Western Lightning ECL (Perkin Elmer) were used according to the manufacturer's instructions. The images were acquired using Chemidoc (Bio-Rad) and analyzed with Imagelab (Bio-Rad), normalizing the intensity value of each fHbp band for the intensity value of the corresponding loading control. Statistical significance compared to the fIR7 was determined using GraphPad Prism 7 software on three independent replicates.

## Serum bactericidal activity (SBA) analysis and survival experiments in human serum

Serum bactericidal activity against *N. meningitidis* strains was evaluated as previously described [61]. Human plasma obtained from volunteer donors under informed consent (MENB REC 2ND GEN-074 (V72_92)) was selected for use as complement source with a particular strain only if it did not significantly reduce CFU of that strain relative to T0 when added to the assay at a final concentration of 50%. The final assay mixture contained 25% human plasma and serial dilutions of mouse anti-fHbp variant 1.1 antiserum. In the human serum survival experiment strains were grown in the same conditions as for the SBA assay, till an $OD_{600}$ of 0.25. Bacteria were diluted to $\sim 10^4$ CFU/ml and 10 μl of the bacterial suspension were added to 190 μl of 60% human serum (Sigma, H4522) in HBSS++ (Sigma). Samples were incubated at 37˚C and 5% $CO_2$, 180 rpm. At various time points (30, 60, 120 and 180 min), an aliquot of each sample was plated in serial dilutions onto GC agar to determine the number of viable bacteria and incubated overnight at 37˚C and 5% $CO_2$. Experiments were performed in triplicate.

## Phylogenetic analysis

The multiple sequence alignment of the 103 sequences from Biagini *et al.* [26] was performed using the MUSCLE algorithm incorporated within the Geneious software (Biomatters) [62]. The matrix of pairwise distances was computed using *seqinr* R package [63], applying *dist.*

*alignment* distance modeling function and considering gaps in the identity measure (gap = 1). The phylogenetic tree was reconstructed using the Neighbor joining method.

### RNA secondary structures analyses

The strength and secondary structure of the Rho-independent terminators were calculated using the online application FindTerm [40] and setting the energy threshold value as -10. Secondary structures and ΔG values of the RNA thermosensor were calculated for 30˚C and 42˚C using the Vienna RNAfold web server [64]. Schematic representations of both terminators and thermosensors were prepared using the Visualization Applet for RNA (VARNA) [65].

### Strain collection from public database

*Neisseria meningitidis* genomes were downloaded from the public Bacterial Isolate Genome Sequence Database (BIGSdb, http://pubmlst.org/software/database/bigsdb/) [43]. The collection included a set of 5818 *Neisseria meningitidis* cases, both from carriage and from invasive disease, collected in the UK before 2016 [66–69]. A perl (Practical Extraction and Report Language) script was created to assign an identification number, the fIR allele, to each unique sequence of the multiple sequence alignment of the intergenic regions downloaded from the BIGSdb. Results of this analysis were linked to the metadata of the corresponding strains present in the BIGSdb (S6 Table). Among the entire set, 5116 strains carried fIRs among to the most common and for which we had a characterization of the expression class (S7 Table). We performed additional analyses only on this subset containing 2139 carrier strains and 2977 invasive strains.

### Statistical analyses

To overcome the bias generated by unbalanced datasets, 1000 datasets have been produced, containing all the 2139 carrier strains and an equal number of invasive strains, randomly selected at each iteration of the algorithm. Logistic regression between fHbp expression groups as predictors and disease status as dependent variable has been performed through the *glm* function from the *stats* package in the R environment. Evaluation of the models' fitting has been performed through *anova* (*stats* package) and *pR2* (*pscl* package), which provides the McFadden index, a pseudo-$R^2$ measure that we used for linear model fitting evaluation. Statistical analyses on the association between fIR alleles and fHbp peptides, variants and clonal complexes were carried out under the R environment. The Pearson residuals [(observed–expected) / sqrt(expected)] were calculated using the *chisq.test* function, which performs chi-squared contingency table tests and goodness-of-fit tests. Visual representation of the analysis was performed using the function *assoc* of the *vcd* R package, that produced an association plot indicating deviations from the independence model in our contingency tables [70].

## Supporting information

**S1 Fig. Multiple sequence alignments of each clade identified in Biagini *et al*. [26].** Within each clade the most representative fIR sequence is indicated on the right. Green areas indicate nucleotides with 100% identity with the most representative fIR allele for the clade, and variable regions are indicated in yellow. Polymorphisms are indicated as Y (pyrimidine, C or T), R (purine, A or G), K (keto, G or T), M (amino, A or C), S (strong, C or G), and N (any nucleotide). Gaps are indicated with "-". At position +128 of the alignment the presence of the ATR insertion element (181 bp sequence, not to scale) in clade III is indicated.
(TIF)

**S2 Fig. Schematic representation of the secondary structures of the Rho-independent terminators.** The structures of the terminators with ΔG = -27.3 kcal/mol **(A)**, -24.7 kcal/mol **(B)**, -14.8 kcal/mol **(C)** and -13.0 kcal/mol **(D)** are represented. SNPs from the strong terminator sequence are highlighted in orange.
(TIF)

**S3 Fig. Summary of capsule groups and clonal complexes present in the public collection of *Neisseria meningitidis* strains.** The heights of the bars refer to the number of strains carrying each capsule group **(A)** and clonal complex **(B)** in the public collection. NA: Not Available; NG: Not Groupable; cnl: capsule null; Various singlets: all strains that have not been assigned to a clonal complex.
(TIF)

**S4 Fig. fIR association to fHbp variants. (A)** In the plot are reported the Pearson's residuals for the 11 most common fIRs in relation to fHbp variants. Statistics have been computed as described in the Materials and methods section. As reported in the legends on the right side, the cells are colored in blue when the number for that combination of alleles is higher than expected by the null hypothesis (that envisions independence), red is used when the number is lower than expected. The darker the color, the bigger the importance of that combination in the *p*-value determination. The height of each bar is proportional to the (signed) residual and the width is proportional to the square root of the expected counts, so that the area of the box is proportional to the difference in observed and expected frequencies. **(B)** Histograms representing number of isolates of each fIR alleles, divided by carrier or invasive disease, in strains harboring fHbp variant 1, variant 2 or variant 3.
(TIF)

**S5 Fig. fIR association to fHbp peptides. (A)** In the plot are reported the Pearson's residuals for the 11 most common fIRs in relation to fHbp peptides. Statistics have been computed as described in the Materials and methods section. As reported in the legends on the right side, the cells are colored in blue when the number for that combination of alleles is higher than expected by the null hypothesis (that envisions independence), red is used when the number is lower than expected. The darker the color, the bigger the importance of that combination in the *p*-value determination. The height of each bar is proportional to the (signed) residual and the width is proportional to the square root of the expected counts, so that the area of the box is proportional to the difference in observed and expected frequencies. **(B)** Histograms representing number of isolates of each fIR alleles, divided by carrier or invasive disease, in strains harboring fHbp peptides 1, 2, 4, 13, 14, 15, 45 or 47.
(TIF)

**S6 Fig. fIR association to clonal complexes. (A)** In the plot are reported the Pearson's residuals for the 11 most common fIRs in relation to clonal complexes. Statistics have been computed as described in the Materials and methods section. As reported in the legends on the right side, the cells are colored in blue when the number for that combination of alleles is higher than expected by the null hypothesis (that envisions independence), red is used when the number is lower than expected. The darker the color, the bigger the importance of that combination in the *p*-value determination. The height of each bar is proportional to the (signed) residual and the width is proportional to the square root of the expected counts, so that the area of the box is proportional to the difference in observed and expected frequencies. **(B)** Histograms representing number of isolates of each fIR alleles, divided by carrier or invasive disease, in strains of clonal complexes ST-11, ST-23, ST-32, ST-41/44, ST-213, ST-269 or ST-461.
(TIF)

**S7 Fig. fIR association to the promoter of** *cbbA***, igr_up_NEIS0350.** In the plot are reported the Pearson's residuals for the 11 most common fIRs in relation to *cbbA* promoters. Statistics have been computed as described in the Materials and methods section. As reported in the legends on the right side, the cells are colored in blue when the number for that combination of alleles is higher than expected by the null hypothesis (that envisions independence), red is used when the number is lower than expected. The darker the color, the bigger the importance of that combination in the *p*-value determination. The height of each bar is proportional to the (signed) residual and the width is proportional to the square root of the expected counts, so that the area of the box is proportional to the difference in observed and expected frequencies. (TIF)

**S1 Table. Values calculated from the box plots in Biagini** *et al.* **[26] and in Fig 1B.** Median, first quartile and third quartile of fHbp amounts plotted according to either the clades or the fIR sequence alleles were extracted from the two box plots. IQR, interquartile range, is calculated as "third—first quartiles" and is an indication of the variability of the set of data. The larger the IQR, the more variable the data set is. (DOCX)

**S2 Table. Polymorphisms that distinguish the 11 fIR alleles identified.** For each of the fIR alleles the characteristics are listed. Definitions of "weak", "medium" or "strong" depend on the free energy prediction of the Rho-independent terminator. The numbers are referred to the position within the multiple sequence alignment (Fig 2E). ATR indicates the presence of the AT-rich insertion element. The sequence of the ribosome binding site (RBS) is underlined. (DOCX)

**S3 Table. List of strains used in this study.** EryR, erythromycin resistance; CmR, chloramphenicol resistance. (DOCX)

**S4 Table. List of primers used in this study.** (DOCX)

**S5 Table. List of plasmids used in this study.** AmpR, ampicillin resistance; CmR, chloramphenicol resistance. (DOCX)

**S6 Table. Dataset used for this study.** (PDF)

**S7 Table. Summary of carrier and invasive isolates for each expression group in the public UK dataset.** The absolute number of isolates in each group is reported along with the percentages for each expression group. (DOCX)

## Author Contributions

**Conceptualization:** Marco Spinsanti, Vincenzo Scarlato, Isabel Delany.

**Data curation:** Marco Spinsanti, Tarcisio Brignoli, Margherita Bodini, Alessia Biolchi.

**Formal analysis:** Marco Spinsanti, Tarcisio Brignoli, Margherita Bodini, Matteo De Chiara, Alessia Biolchi, Alessandro Muzzi.

**Funding acquisition:** Isabel Delany.

**Investigation:** Marco Spinsanti, Tarcisio Brignoli, Margherita Bodini, Lucia Eleonora Fontana, Matteo De Chiara, Alessia Biolchi, Alessandro Muzzi, Vincenzo Scarlato, Isabel Delany.

**Methodology:** Marco Spinsanti, Tarcisio Brignoli, Margherita Bodini, Lucia Eleonora Fontana, Matteo De Chiara, Alessia Biolchi.

**Software:** Marco Spinsanti, Margherita Bodini, Alessandro Muzzi.

**Supervision:** Alessandro Muzzi, Vincenzo Scarlato, Isabel Delany.

**Validation:** Marco Spinsanti, Tarcisio Brignoli, Margherita Bodini, Lucia Eleonora Fontana, Alessia Biolchi.

**Visualization:** Marco Spinsanti.

**Writing – original draft:** Marco Spinsanti, Isabel Delany.

**Writing – review & editing:** Marco Spinsanti, Tarcisio Brignoli, Margherita Bodini, Lucia Eleonora Fontana, Matteo De Chiara, Alessia Biolchi, Alessandro Muzzi, Vincenzo Scarlato, Isabel Delany.

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
