## [Decision Letter · Decision Letter 0]

30 Oct 2020

Dear Dr. Delany,

Thank you very much for submitting your manuscript "Deconvolution of intergenic polymorphisms determining high expression of Factor H binding protein in meningococcus and their association with invasive disease" for consideration at PLOS Pathogens. As with all papers reviewed by the journal, your manuscript was reviewed by members of the editorial board and by several independent reviewers. In light of the reviews (below this email), we would like to invite the resubmission of a significantly-revised version that takes into account the reviewers' comments. 

You will note that all three Reviewers expressed the need to improve the clarity of presentation of the manuscript and to reduce speculation; I agree with these editorial concerns.  Importantly, there is concern regarding Figure 6 and presentation of the cropped version of the western blots (6A) as well as documentation of loading controls. Lastly, it is essential that you verify the qRT-PCR with transcript levels from a second control gene,  Thus, some additional experiments may be required to correct these issues.

We cannot make any decision about publication until we have seen the revised manuscript and your response to the reviewers' comments. Your revised manuscript is also likely to be sent to reviewers for further evaluation.

Sincerely,

William M Shafer, Ph.D.

Guest Editor

PLOS Pathogens

Xavier Nassif

Section Editor

PLOS Pathogens

Kasturi Haldar

Editor-in-Chief

PLOS Pathogens

orcid.org/0000-0001-5065-158X

Michael Malim

Editor-in-Chief

PLOS Pathogens

orcid.org/0000-0002-7699-2064

Reviewer's Responses to Questions

**Part I - Summary**

Reviewer #1: (No Response)

Reviewer #2: The manuscript "Deconvolution of intergenic polymorphisms determining high expression of Factor H binding protein in meningococcus and their association with invasive disease" by Spinsanti et al. is a molecular and epidemiological study on the production level of Factor H binding protein and their association to reported meningococcal invasive cases.

The overall principle and idea of the work is good, however, the authors did not show convincing data to support on the mechanism of FHbp production, the epidemiological section is unsupported and the conclusion in association to invasive cases is too bold and not supported.

Reviewer #3: The manuscript entitled “Deconvolution of intergenic polymorphisms determining high expression of Factor H binding protein in meningococcus and their association with invasive disease” by Spinsanti et al. extends previous studies of the promotor region of FHbp. The authors identify fHbp intergenic regions (fIRs) that were previously referred to as “clades” by Biagini et al. (PNAS, 2016, 113(10): 2714). The authors perform a detailed analysis of the major allele sequences within each fIR and correlate fHbp expression levels (grouped into 4 categories) with the prevalence of the corresponding fIR among carriage and disease isolates from the U.K. The experimental approaches are sound, however the organization and presentation of the manuscript could be improved.

**Part II – Major Issues: Key Experiments Required for Acceptance**

Reviewer #1: (No Response)

Reviewer #2: Major issues.

1) As the authors mentioned and shown in figure 1A, the expression of fHbp is dependent on a bicistronic and monocistronic transcript. While it is interesting to speculate polymorphisms within the intergenic region between the first gene cbbA and fHbp contribute to the production of FHbp, the authors have neglected to investigate the polymorphism within the promoter region of cbbA. Work by Oriente et al., and Delaney in J. Bact (2010) has previously shown that the major expression of fHbp and the production of FHbp originate from the bicistronic mRNA transcript. With cbbA (nmb1869) deleted, the expression of fHbp is down to perhaps 10%.

Next, the authors identified several polymorphisms within the cbbA-fHbp intergenic region and grouped them into different classes (fIR). The authors then generated isogenic mutants in a WT cbbA promoter with these fIRs and showed that the FHbp were expressed differently. It is unclear whether the promoter region of cbbA would have massive affect the expression of fHbp. To confirm this, the authors need to show polymorphisms within the cbbA promoter regions to the basal expression of fHbp and their association to the polymorphisms within the fIRs. Example, if there are several known polymorphisms within the promoter of cbbA, and the fIR is in WT configuration, what is the basal level of FHbp production? Are the specific polymorphisms in the promoter of cbbA associated to the fIR polymorphisms? The best way for authors to show this is to have the fIRs in a cbbA mutant background used by them in Oriente et al. J. Bact (2010).

2) Figure 2B,C,D. qRT-PCR on the RNA transcript levels are not convincing and there was only 1 loading control using 16S RNA (minimum of 2 loading controls are required). The authors should perform Northern blots instead to show that the effect seen on FHbp is not due to stability of the bicistronic mRNA.

3) What is the loading control in Figure 3B? It was neither mentioned in the Figure legend nor in the material and methods. The signal for this loading control is extremely weak. Please repeat this blot and consider showing a stronger signal loading control (perhaps CbbA). Is this Western blot repeated or was it just performed one time?

4) Figure 6A is fully not acceptable even though it is used to show an example level of FHbp which was later quantified in 6B. The author should never crop western blot bands and place them in boxes next to each other. The authors should consider repeating the gel and load them in the proper sequential manner to convince the reader. There are also no loading controls, how do the authors know the various fIRs were not loaded differently to show the production of FHbp? It is also not clear what the quantification of the band intensities was ratio to in Fig 6B.

5) Figure 6C and D. It is unclear human plasma from volunteer donors and the human serum (Sigma) has been verified to contain meningococcal FHbp antibody. The authors should perform Western blots using FHbp variant antibodies on the serum to show that the serum used in Fig6D, does not harbour specific antibody/antibodies towards FHbp different variants. Otherwise, the serum survival results can be skewed and is unreliable. I.e. how many of the subjects had prior vaccination against Bexsero or recent colonisation/infection of meningococci? The author should also raise this issue in the discussion.

6) Figure 7 (epidemiological part) claiming that fIRs with high, medium high to low production of FHbp is not convincing and difficult to understand. It is understandable that the authors correlate the higher production of FHbp would enable to the bacterium survival better in the host with immune evasion thus leading the manifestation of meningococcal invasive diseases. However, the epidemiological data do not add up.

There are several major points the authors need to investigate and raise prior to making such conclusions.

A) Invasive isolates isolated from year 2010-2012 and carriers from year 1996-2015. There is a bias in this sampling due to several factors and a few big outbreaks can influence the variety of invasive isolates obtained from 2010-2012. There is an attempt to deal with this in the “Statistical analyses” however, in the methods used; it is unclear exactly how this bias is being dealt with. Creating 1000 datasets with random sampling from the invasive isolates will still contain isolates with a great dependency on possible outbreaks, making them still potentially very similar and biased – just in a 1000 very similar ways.

B) Among these 1100 strains (should be renamed to isolates and not strains) from the UK, how many of them are of various capsule group/CC types also have effect on the fIRs of association to invasive cases? The supplementary figure showing number of isolates of different serogroups and CC is not informative.

C) The majority of the isolates are of sergroup B (under capsule) and ST-41/44 (under CC)and if one looks into the supplementary data, ST-41/44contained up to 40% of fIR2 (medium low FHbp production), how is this even relate to the high association with invasive cases? How many isolates of serogroup B and are of ST-41/44 too?

D) ST-11, serogroup W is a rising “clade” of hypervirulent strains responsible for a big part of meningococcal infections, although as presented in this data is one of the lowest (fIR4) production of fHbp – Data shows that the invasiveness properties of fIR variants seems to be more attributed to specific ccs such as cc32 and cc269. Please investigate and report

E) Why is ST-22 not shown with the distribution of fIR?

F) Under the statistical analyses, the authors used several packages such as pR2 and vcd R. Please elaborate on what they do to the dataset.

Reviewer #3: 1. It is not clear how exactly what criteria were used to define the four classes of expression were determined. The text on page 14, line 329, indicates protein expression (in isogenic mutant strains) and the data in Figure 6 show the clearest distinction in susceptibility to bactericidal activity of the same strains to anti-fHbp antiserum.

2. Figure 7. It would be helpful to indicate how many strains used in the analysis were in each expression group (and within each expression group, N for each carriage and invasive). Also, the logistic regression suggests that more disease is caused by isolates with medium high expression whereas the data show that the opposite is true. Could this an artifact of a small sample size for some of the expression groups or trying to fit data for which there might be many other variables (strain clonal complex, fHbp sequence variant, etc.)?

**Part III – Minor Issues: Editorial and Data Presentation Modifications**

Reviewer #1: (No Response)

Reviewer #2: Minor issues

1) Figure 4 and 5 are wrongly labelled

2) Please use unit for the Delta G values (Lines 227, 228)

3) Line 242. Authors claim that the ribosome binding site (AGGAG) would affect translatability. Please show the result otherwise the tone down the claim.

Reviewer #3: 1. Figures. Figure 1B and C contain significant overlap with Biagini et al. (data replotted with a different nomenclature). Fig. 1A is not very helpful and duplicates with Figure 2A. Figs 1A and D could be combined to show the promotor elements and the polymorphisms could be indicated, as they already are, in Fig. 2E

2. Organization of text. First paragraph of Results belongs elsewhere (Introduction). Paragraph beginning on page 10, line 212 should be moved to the Discussion section as it is somewhat speculative. Elsewhere, there excess discussion with Results sections. Overall, text in Results and Discussion is lengthy.

3. Page 11, line 249. Regarding the weak vs. strong terminator hypothesis, the text says “sequence of wild-type fIR intergenic region was modified with the substitution of a single nucleotide within the stem region.” Figure 3 seems to indicate that this substitution was in the background of the -10 box substitutions, not the WT background. Also, line 257, description of Figure 3: this section would be more clear if the text were written to follow the sequence in the Figure or vice versa.

4. The abstract mentions a correlation between FHbp expression and survival. This could be assessed directly by plotting the data in Figure 6C versus the data in 6B. Have the authors performed this analysis and is there a significant correlation?

5. Figures 4 and 5. Order of Figures does not match text.

PLOS authors have the option to publish the peer review history of their article (what does this mean?). If published, this will include your full peer review and any attached files.

Reviewer #1: No

Reviewer #2: No

Reviewer #3: No
---

## [Editor Report · Decision Letter 1]

9 Mar 2021

Dear Dr. Delany,

We are pleased to inform you that your manuscript 'Deconvolution of intergenic polymorphisms determining high expression of Factor H binding protein in meningococcus and their association with invasive disease' has been provisionally accepted for publication in PLOS Pathogens.

Best regards,

William M Shafer, Ph.D.

Guest Editor

PLOS Pathogens

Xavier Nassif

Section Editor

PLOS Pathogens

Kasturi Haldar

Editor-in-Chief

PLOS Pathogens

orcid.org/0000-0001-5065-158X

Michael Malim

Editor-in-Chief

PLOS Pathogens

orcid.org/0000-0002-7699-2064

Thank you for your responsive changes to the manuscript and your attention to detail. Based on the edits and your responses as well as my overall evaluation, I decided not to send the paper back to the Reviewers.
---

## [Editor Report · Acceptance letter]

23 Mar 2021

Dear Dr. Delany,

We are delighted to inform you that your manuscript, "Deconvolution of intergenic polymorphisms determining high expression of Factor H binding protein in meningococcus and their association with invasive disease," has been formally accepted for publication in PLOS Pathogens.

Best regards,

Kasturi Haldar

Editor-in-Chief

PLOS Pathogens

orcid.org/0000-0001-5065-158X

Michael Malim

Editor-in-Chief

PLOS Pathogens

orcid.org/0000-0002-7699-2064